# Targeting the Host Mitochondria as a Novel Human Cytomegalovirus Antiviral Strategy

**DOI:** 10.3390/v15051083

**Published:** 2023-04-28

**Authors:** Lauryn O. Bachman, Kevin J. Zwezdaryk

**Affiliations:** 1Department of Cell and Molecular Biology, Tulane University School of Science and Engineering, New Orleans, LA 70112, USA; 2Department of Microbiology and Immunology, Tulane University School of Medicine, New Orleans, LA 70112, USA; 3Tulane Brain Institute, Tulane University School of Medicine, New Orleans, LA 70112, USA; 4Tulane Center for Aging, Tulane University School of Medicine, New Orleans, LA 70112, USA

**Keywords:** cytomegalovirus, CMV, mitochondria, ETC, metabolism, antivirals

## Abstract

Human cytomegalovirus (HCMV) exploits host mitochondrial function to promote viral replication. HCMV gene products have been described to directly interact and alter functional or structural aspects of host mitochondria. Current antivirals against HCMV, such as ganciclovir and letermovir, are designed against viral targets. Concerns with the current antivirals include toxicity and viral resistance. Targeting host mitochondrial function is a promising alternative or complimentary antiviral approach as (1) drugs targeting host mitochondrial function interact with host targets, minimizing viral resistance, and (2) host mitochondrial metabolism plays key roles in HCMV replication. This review describes how HCMV alters mitochondrial function and highlights pharmacological targets that can be exploited for novel antiviral development.

## 1. Introduction

Viruses are obligate parasites and require remodeling of host pathways for efficient viral replication. Hijacking host metabolic pathways permits production of nucleic acids, proteins and lipids required for viral replication. Interfering with host metabolism should directly impact the virus, implicating metabolism as a key factor influencing viral infection and replication. Changes to glycolysis, oxidative phosphorylation (OXPHOS), fatty acid synthesis, glutaminolysis and other metabolic pathways have been characterized during infection and replication by many diverse viruses (reviewed in [1,2,3]). As viruses differ in structure, genetic backbone, and site of replication, diverse metabolic strategies have emerged, dependent on rate of replication and viral size. Reprogramming of metabolic pathways can impact immune response, apoptosis, and establish metabolically favorable conditions for efficient viral replication.

Human cytomegalovirus (HCMV) is a common herpesvirus exhibiting 40–80% seroprevalence. Infection in immunocompetent individuals is typically asymptomatic, but as with all herpesvirus infections, the virus is never cleared and enters a latent phase. Periodic reactivations occur throughout a lifespan and increase in frequency with age. Under immunosuppressed conditions, HCMV infection or reactivation may result in more severe pathology. This includes organ rejection and increased morbidity and mortality under transplantation settings. HCMV is also the leading viral cause of congenital infections.

HCMV is a master manipulator of host cellular metabolism [4]. Upon infection, HCMV upregulates glycolysis and glutaminolysis [5,6,7,8,9,10,11,12]. Metabolites derived from glycolysis and glutaminolysis enter the tricarboxylic acid (TCA) cycle, altering OXPHOS and electron transport chain (ETC) function [5,8,13,14,15,16]. A simultaneous increase in glycolysis and OXPHOS is reported in cancer stem cells and has been observed in diverse viral classes. The elevated carbon flux in the TCA cycle has been shown to support fatty acid synthesis [10,17,18,19]. The inhibition of host mitochondrial pathways negatively impacts HCMV replication [20]. An in-depth description on how metabolism is altered during HCMV replication and how HCMV manipulates host mitochondrial function can be found elsewhere [21,22]. Together, this suggests HCMV dependence on the host mitochondria for successful HCMV infection and replication. This also provides a target for development of novel antivirals. 

Mitochondria are also associated with apoptotic pathways [23]. Apoptosis is a type of programmed cell death that is associated with decreased mitochondrial membrane potential (reviewed in [24]). HCMV has been described to prevent apoptosis through numerous mechanisms [25]. Specifically, HCMV can inhibit intrinsic apoptosis by targeting the Bcl-2 family and associated proteins located at the mitochondria [26,27]. This review includes intrinsic apoptosis pathways when related to changes in mitochondrial function. Further reading on the interplay between HCMV and apoptosis, can be found here [28,29].

HCMV antivirals are efficient and have a long history of clinical success. However, current drugs disrupt viral targets making them susceptible to viral resistance. By identifying and optimizing host targets that are essential for HCMV replication, viral resistance can be avoided. Targeting the host mitochondria as an antiviral could be utilized effectively as an adjunct treatment with current HCMV antivirals to reduce treatment duration and increase antiviral efficacy.

In this review, we discuss how HCMV alters host mitochondria function to enhance replication. We examine current and emerging evidence that shows the feasibility of targeting host mitochondria and how pharmacological compounds impact HCMV replication and mitochondrial or cellular health. Throughout this review, we provide examples specific to HCMV but which are likely applicable to many viral classes. As HCMV infection and cancer exhibit similar metabolic changes, we explore novel mitochondria-targeted approaches in the cancer field and apply the lessons learned to develop these treatments as antivirals.

## 2. HCMV Targets Host Mitochondria

There are five HCMV products currently defined to interact with the host mitochondria (Figure 1). These are HCMV pUL37x1, long non-coding β2.7 (lncβ2.7), pUL13, pUL15A, and US9 [13,14,30,31,32,33]. Strong evidence supporting the essential role of host mitochondria during HCMV replication was provided by infecting Rho cells with HCMV [8]. Human foreskin fibroblasts were used to generate Rho cells that are characterized as having depleted mitochondrial DNA (mtDNA). Rho cells do not have a functional ETC, but through media supplementation, maintain functional mitochondria. The Rho cells displayed 90–95% knock down of mtDNA as determined by ND1 (a mtDNA gene) expression. HCMV-infected Rho cells exhibited a 90% decrease in viral titers, emphasizing the importance of a functional ETC during HCMV infection and replication. Unsurprisingly, the HCMV gene products identified to interact with host mitochondria predominantly interact with the ETC function and/or OHPHOS. HCMV UL37x1 encodes viral mitochondria-localized inhibitor of apoptosis (pUL37x1 or vMIA) that has been described to bind the ETC ATPase inorganic phosphate carrier (PiC). This interaction decreases phosphate transport, resulting in lower ATP production [34]. pUL37x1 has been reported to promote mitochondrial membrane stability [35]. Further, HCMV UL37x1 knockout strains were reported to decrease mitochondrial respiration compared to wild-type strains, suggesting increased mitochondrial respiration is in part due to pUL37x1 [14]. pUL13 targets and remodels cristae, directly interacts with mitochondrial proteins inner membrane mitochondrial protein (IMMT) and coiled-coil-helix-coiled-coil-helix domain containing 3 (CHCHD3) and increases mitochondrial respiration [13]. HCMV US9 targets and disrupts mitochondrial membrane potential and integrity by interfering with translocase of outer membrane 20 (TOM20) and TOM70 [36]. Disruption of the membrane potential releases mitochondrial antiviral signaling protein (MAVS) from the outer mitochondrial membrane [36]. The disruption to MAVS is postulated to be a virus-induced immune evasion strategy. The mitochondrial targets of pUL15A are currently unknown, but localization with host mitochondria was observed late in infection (96 h post infection) [31].

Other HCMV products such as HCMV-miR-UL36-5p miRNA may target the mitochondria, but conclusive data are lacking. HCMV-miR-UL36-5p interacts with adenine nucleotide translocator 3 (ANT3), also termed SCL25A6, to inhibit cell death by decreasing ANT3 expression [37]. ANTs have been implicated to have roles in the mitochondrial permeability transition pore (mPTP). Interestingly, lncβ2.7 and pUL13 are expressed early during the HCMV infection and replication cycle, suggesting mitochondrial control and/or stability is a priority during HCMV replication [32,38].

## 3. ETC and Its Role in HCMV Replication

The ETC consists of transmembrane protein complexes embedded in the cristae, located in the mitochondrial inner membrane. Ubiquinone and cytochrome c (Cyt c) allow electron transfer between complexes. An in-depth review of ETC function is detailed elsewhere [39]. HCMV has been reported to increase ETC activity at transcriptional and protein levels [8,13,15,40]. An overview of the effect of mitochondrial disruptions on HCMV replication is provided in Table 1.

### 3.1. ETC Complex I: NADH-Ubiquinone Oxidoreductase

Complex I transfer electrons from the matrix NADH to ubiquinone. Electrons entering the ETC are transferred by iron–sulfur clusters to ubiquinone (Coenzyme Q10). The transfer of a pair of electrons in complex I (NADH to CoQ) induces translocation of protons from the matrix to the intermembrane space. Complex I may be of critical importance during HCMV replication as it regenerates oxidized NAD^+^ and is the entry point for electrons into the respiratory chain. Mutations of complex I subunits result in numerous clinical presentations including leukodystrophy, optic neuropathy, and encephalopathy (reviewed in [41,42]). Interestingly, leukodystrophy and congenital CMV are difficult to distinguish [43]. 

Targeting complex I of the ETC has shown success in mitigating CMV replication [20,32]. The addition of rotenone, a natural isoflavonoid and irreversible inhibitor of complex I, significantly decreased HCMV viral titers three days post infection [32]. Using U737 cells, rotenone addition during HCMV infection increased oxidative stress and decreased ATP production. The HCMV 2.7-kb lncβ2.7 was reported to colocalize with complex I, modulating metabolic intermediates [32]. lncβ2.7 was found to bind GRIM19, protecting infected cells from metabolic-stress-associated apoptosis. Infection with a HCMV lncβ2.7 knockout strain resembled rotenone-treated cells including changes to metabolism, highlighting a possible interaction between complex I and HCMV [32]. Cell cytotoxicity due to rotenone use was not significant. Further, HCMV lncβ2.7 stabilized mitochondrial membrane potential and maintained ATP production in infected cells. Significant decreases in mitochondrial membrane potential were measured when cells were infected with a knockout lncβ2.7 strain. This was similar to treating cells two hours post HCMV infection with a low dose of rotenone (0.1 µM), which reduced HCMV viral titers by 1 log at day 6 post infection [20]. Significant decreases in viral titer were reported beginning at day 4 post infection. Interestingly, expression of the viral protein immediate-early 1 (IE1) was unaffected by rotenone treatment, but expression of early (pp52) and late (pp28) viral proteins were delayed or reduced. Rotenone did decrease cell viability, potentially impacting results. 

Metformin has been described to inhibit complex I or mediate metabolic changes through mTOR [44]. Preliminary studies show a single metformin treatment, 2 h post HCMV infection, decreased HCMV titers by 1 log [20]. When glucose levels in the culture medium were restricted (1 g/L versus 4.5 g/L), a 2.5 log reduction in HCMV titers was observed at day 6 post infection. The trends displayed reduced viral titers beginning at day 3 post infection and delayed expression of HCMV early and late proteins. HCMV-infected cells treated with metformin exhibited increased viability although the change was not significant. It is interesting to speculate on the efficacy of metformin if treatment was to be administered daily rather than a single dose shortly after initial infection.

### 3.2. ETC Complex II: Succinate Dehydrogenase 

Complex II bridges metabolism and OXPHOS [45]. Complex II induces the oxidation of succinate to fumarate and is an entry point for electrons. Through iron–sulfur clusters, electrons from succinate are transferred to ubiquinone, permitting reduction to ubiquinol. Malonate is a three-carbon dicarboxylic acid and is considered a competitive inhibitor of complex II succinate dehydrogenase [46]. Diazoxide (DZX) is a mKATP-channel agonist that is known to inhibit complex II [47]. Malonate administration acts as a complex II inhibitor by mimicking DZX and decreasing ROS [48]. Although no direct experiments involving HCMV and ETC complex II have been reported, HCMV has demonstrated the ability to induce ROS production as infection persists [33,49]. Therefore, malonate’s inhibition of complex II and downregulation of ROS suggests that overexpressing malonate could inhibit the progression of the ETC and impact HCMV viral replication. This poses a potential target for future studies but has yet to be explored.

### 3.3. ETC Complex III: CoQ-Cytochrome c Reductase

Complex III transfers electrons from ubiquinol to Cyt-c. This reaction takes up two protons from the mitochondrial matrix and transfers four protons to the other side of the mitochondrial membrane. Antimycin A, an ETC complex III inhibitor, prevents electrons from transferring from Complex I or FADH2 to Cyt-c [50]. During HCMV infection, a single dose of Antimycin A given 2 h post infection resulted in a 2-log reduction in HCMV titers using HCMV TR, but only a 0.5-log change using HCMV Towne [20]. There were no significant changes in cell death associated with cytotoxicity. A mechanism to describe this difference between strains is unknown but suggests that HCMV laboratory strains may exhibit greater resistance to metabolic stress under in vitro conditions as titers were significantly decreased at day 3 post infection. Complimentary experiments involving daily treatments with Antimycin A would provide insight to these observations. Of note, when released from the mitochondria, Cyt-c plays a role in apoptosis by shuttling electrons from oxidative phosphorylation complex III to IV [51].

### 3.4. ETC Complex IV: Cytochrome c Oxidase

Complex IV generates H_2_O by transferring electrons from Cyt-c to a terminal electron acceptor (O_2_)_._ An experimental design specifically targeting complex IV during HCMV infection is lacking, but studies using rubella virus described a decrease in complex IV activity [52]. Alternately, HIV infection of a CD4^+^ T cell line increased complex IV activity [53]. The changes were driven by virus-induced apoptosis as inhibition of complex IV using potassium cyanide reduced apoptosis using the same CD4^+^ T cell model. The role of complex IV during HCMV infection is currently a gap in knowledge that should be addressed.

### 3.5. Complex V: F_1_F_0_ ATPase Synthase

Complex V phosphorylates ADP to ATP. Electron transfer to monooxygen generates H_2_O and the pumping of protons from the matrix to the inner membrane space (via Complex I, III and IV as described above). Protons pass from the inner membrane space to the matrix transferring stored energy generated by the proton electrochemical gradient from F_0_ to F_1_. The induces a conformational change in F_1_F_0_ synthase, permitting ATP formation. 

Oligomycin differs from other ETC inhibitors because it targets the proton channel of ATP synthase as opposed to directly inhibiting the ETC [50]. Oligomycin binds to a subunit of ATP synthase, preventing protons from passing back into the mitochondria and creating an unfavorable proton gradient for its operation [50,54]. The TCA cycle cannot adequately operate under these conditions, because NADH remains high and NAD^+^ is too low [54]. By inhibiting ATP synthesis in the mitochondrial matrix, oligomycin limits HCMV replication [20,55]. HCMV-infected cells were more sensitive to the effects of oligomycin than mock-infected cells [14]. This suggests that HCMV-infected cells exhibit greater respiration efficiency and may tolerate greater bioenergetic stress. Oligomycin resistance was more pronounced in cells infected with a HCMV strain lacking UL37x1 compared to the parental strain [14]. The increased susceptibility to the effects of oligomycin in HCMV-infected cells could indicate that pUL37x1 interacts with oligomycin. Further, the increase in respiration activity during HCMV infection was partially dependent on pUL37x1 [14]. Further experiments with HCMV pUL37x1 knockout strains suggest that expression is necessary for HCMV replication and the inhibition of apoptosis [56]. A single treatment of oligomycin after HCMV infection reduced viral titers by 1 log or resulted in no change at day 6 post infection using HCMV TR and Towne, respectively [20]. This result was not influenced by cytotoxicity.

### 3.6. ETC-Associated Mechanisms

The mitochondrial membrane potential has been reported to be hyperpolarized [8,32,57,58] or depolarized [36,59] following HCMV infection. Chloramphenicol uncouples the ETC and treatment was reported to reduce HCMV titers, but this was not driven by changes in respiration [14]. Addition of the uncoupling compound carbonyl cyanidetrifluoromethoxyphenylhydrazone (FCCP) did delay HCMV replication at day 3 post infection but did not significantly change HCMV viral titers at day 6 post infection [20]. Again, this was single dose, shortly after initial HCMV infection. The arteminisinin-derived compound BG95 has recently been shown to colocalize with the mitochondria and reduce membrane potential [60]. 

Other artemisinin derivatives, including the semisynthetic analogue artesunate, have been shown to be effective antivirals against HCMV [61,62,63]. Two bioactive excipients, poloxamer 188 and quercetin, were co-formulated to yield Quercetin-P188. When administered with ganciclovir, Quercetin-P188 was shown to synergistically enhance the efficacy of ganciclovir [64]. Interestingly, it was found to colocalize with mitochondria and P188 has been reported to protect mitochondria. Hyperpolarization has been reported to occur during mitochondrial stress and is accompanied with elevated levels of ROS [65,66]. Increased mitochondrial membrane potential and ROS, specifically mitochondrial-derived superoxide, has been reported during CMV infection [8,67]. Increased ROS is associated with inflammasome assembly and the production of the pro-inflammatory cytokine IL-1β. Experimental designs linking OXPHOS, mitochondrial membrane potential, ROS and inflammasome assembly are yet to be explored.

## 4. Mitochondrial Morphology and Its Role in CMV Replication

Mitochondrial fission and fusion are processes that maintain mitochondrial integrity. This cycle allows the transfer of gene products amongst mitochondria, ensuring optimal functioning. Stress or disruption of this cycle can contribute to various diseases. 

Fragmented mitochondria are observed following HCMV infection [8,27,30,34,59]. HCMV pUL37x1 promotes mitochondrial fission by elevating Ca^2+^ levels in the mitochondria through induction of Ca^2+^ from the ER to the cytosol [30,35,58,68,69]. Dynamin-related protein 1 (Drp1) regulates fission and is recruited to the mitochondria during Ca^2+^ flux. Excess mitochondrial fission is often associated with the first steps of apoptosis [69]. Further, pUL37x1 was observed to recruit Bax, an apoptosis-inducing protein, to the mitochondria, disrupting the permeability of the outer mitochondrial membrane [70]. The fragmentation of mitochondria was also observed in DU145 cells and HCT116 Bax^−/−^ cells [71]. These observations suggest pUL37x1 could be a potential drug target [72]. pUL37x1 disruptions to the mitochondria could potentially be inhibited with a Ca^2+^ chelating agent [69]. Changes in mitochondrial architecture and HCMV mechanisms underlying these changes have been elegantly shown to be dependent on protein posttranslational modifications [73,74]. HCMV pUL13 was shown to localize within the mitochondria by 48 h post infection [13]. Interactions between pUL13 and cristae-shaping proteins altered the mitochondrial architecture, increasing cellular respiration. Specifically, pUL13 interaction with the mitochondrial contact site and cristae organizing system (MICOS) was observed. Remodeling of the inner mitochondrial membrane occurs through interactions with the protein complex MICOS, ATP synthase and inner membrane phospholipids. Poor maintenance or stability of cristae membrane can impair mitochondrial function. pUL13 appears to stabilize MICOS as a mechanism to promote mitochondrial function during a period of extreme mitochondrial demand and stress. 

## 5. Prospective HCMV Antivirals and Respective Mode of Action

Many of the ETC targets outlined above (rotenone, Antimycin A) exhibit systemic toxicity. Numerous therapeutics have been developed, largely in the cancer field, for specific targeting to the mitochondria (Table 2). In this section we introduce drugs that could potentially be tested as antivirals. Tamoxifen, α-tocopheryl succinate (α-TOS), 3-bromopyruvate, and metformin have been reported to inhibit ETC function, increase ROS and contribute to increased cancer cell death (reviewed in [75,76]). Mitochondria-targeted Tamoxifen (MitoTam) showed efficient induction of cell death using breast cancer-related Her2^high^-expressing cells [77]. MitoTam inhibited ETC supercomplex assembly, elevated ROS production and promoted cell death. The mitochondrial membrane potential was reduced in cells, suggesting that complex I of the ETC is responsible for the observed changes. The synthetic isoflavin analogues ME-143 and ME-344 target complex I and induced apoptosis in various cancer cell lines [78]. Both compounds work similarly to rotenone. ME344 interferes with ETC complex subunits and both compounds decrease mitochondrial membrane potential [78]. The proposed mechanism involved induction of apoptosis due to prolonged decreases in mitochondrial membrane potential. A vitamin E subgroup, γ-tocotrienol, interacts with ETC complex I and II subunits, increasing ROS, resulting in apoptosis in two cancer cell lines [79]. A second drug, mitochondria-targeted analog of vitamin E succinate (MitoVES), was shown to rapidly induce ROS and drive cell death through a Bcl-2-mediated mechanism [80]. ETC complex II is the proposed target of MitoVES, which interferes with ubiquinone function and mitochondrial membrane potential [81]. MitoVES was shown to selectively induce apoptosis in malignant cells and suppress tumor progression [80]. Collectively, pharmacological targeting of the ETC, induces apoptosis of cancer cells.

Pairing drugs that target different metabolic pathways is an advanced approach that has shown promising results. Using a xenograft model of human breast cancer, therapeutics targeting mitochondria, such as mitochondria-targeted carboxyl-proxyl (Mito-CP) or MitoQ, in combination with the glycolysis inhibitor 2-deoxy-D-glucose (2-DG) resulted in tumor regression [82]. 

Other drugs, such as the cyclopamine analogue cyclopamine tartrate (CycT), a hedgehog (Hh) signaling inhibitor, disrupted mitochondrial function in lung cancer cell lines [83]. Again, elevated ROS led to increased cell death. Interestingly, mitochondrial membrane potential was increased and mitochondrial fragmentation was observed, similar to observations reported during HCMV infection of fibroblasts [8]. CycT was also shown to disrupt OXPHOS and heme metabolism [84]. Heme is critical for efficient mitochondrial respiration, specifically ETC complexes II–IV. 

Metformin has been described to alter different metabolic targets including mTOR and ETC complex I [85,86,87]. Treating cancer cells with metformin reduces TCA cycle intermediates and short chain acyl carnitines, depletes ADP and ribonucleotide and deoxyribonucleotide triphosphates, and increases aerobic glycolysis [88,89,90,91]. Some of the described changes were the result of metformin interacting with ETC complex I. 

A quinazolinone derivative, the mitochondrial DIVision Inhibitor 1 (mDIVI1) efficiently increased cell death using cancer cell lines in vitro [92]. mDIVI1 inhibited the mitochondrial fission protein DRP1, decreasing mitochondrial metabolism and increasing ROS. It is interesting to speculate about the importance of ROS. Data have shown significantly high levels of ROS correlate with the increase in cell death during HCMV infection [83]. A recent study concluded that HCMV lncβ2.7 assisted in the upregulation of superoxide dismutase 2 (SOD2), maintaining acceptable levels of ROS, thus protecting HCMV-infected monocytes from apoptosis [33]. 

ME 143, ME 344, α-tocopheryl and metformin have completed or are currently in clinical trials as cancer therapeutics. Metformin and α-tocopheryl reported low adverse events, suggesting that toxicity issues are not evident. ME 143 and ME 344 did report numerous adverse events (including lactic acidosis), suggesting that these drugs may need reformulation before further clinical testing. There are numerous other approaches undertaken to exploit mitochondrial metabolism as a target for cancer therapy (reviewed in [93,94,95]). A list of clinical trials, targeting mitochondrial metabolism as a therapeutic in cancer is reviewed here [96].

**Table 2 viruses-15-01083-t002:** Mitochondrially targeted drugs defined for cancer.

Drug	Target	Results	Notes	Reference
MitoTam	ETC complex I	- Increased ROS - Decreased mitochondrial membrane potential	No change in glycolysis or ATP levels	[77]
ME-143 ME-344	ETC complex I	- Decreased mitochondrial membrane potential	ME-344 induced ETC complex disruption	[78]
γ-tocotrienol	ETC complexes I and II	- Increased ROS		[79]
MitoVES	ETC complex II (proposed)	- Increased ROS - Decreased mitochondrial membrane potential	Decreased effect on non-malignant cells	[80]
CycT	OHPHOS	- Increased mitochondrial membrane potential - Increased ROS - Increased mitochondrial fission		[83]
		- Decreased OXPHOS and expression of ETC subunits	Independent of hedgehog signaling	[97]
Metformin	ETC complex I	- Increased glycolysis - Uncoupled respiration		[91]
		- Decreased ETC function	Requires increased MMP	[90]
	Not described	- Decreased TCA intermediates		[89]
		- Decreased TCA intermediates and carnitines	Increased aspartate release	[88]
mDIVI1	DRP1	- Increased ROS		[92]

## 6. Conclusions and Future Perspectives

The new therapeutic strategies described in this review still require thorough analysis on specificity, dosage, toxicity and efficacy prior to moving forward. The list of pathologies associated with HCMV continues to expand, despite the lack of mechanistic understanding. Glioblastoma is an ideal example. The presence of HCMV nucleic acids or proteins in glioblastoma remains controversial, but data suggest that treating patients diagnosed with glioblastoma with the antiviral valganciclovir nearly doubles the median overall survival [98]. Novel strategies to reduce HCMV viremia or viral burden efficiently and efficaciously could benefit patient care outside of individuals normally associated with HCMV susceptibility or disease.

The use of mitochondria-targeted antivirals would ideally be used synergistically with current antivirals. It is expected that coadministration of antiviral and mitochondria- targeted drugs would reduce the duration that patients require antiviral treatment. Theoretically, this would diminish the probability of antiviral resistance occurring and minimize toxicity issues due to prolonged antiviral use. Again, the efficacy and safety studies must be completed to validate this strategy. Collectively, this would improve patient care and ease the financial burden of the patient and to the healthcare system. With continued clinical evidence suggesting HCMV can impact the outcome of patients with cardiovascular disease [99,100], cancer [98,100] or dementia-related disease [97], the use of novel mitochondria-targeted antivirals could provide multiple benefits. A single therapeutic designed to target host mitochondrial pathways could provide benefits by (1) reducing the negative impact of localized or systemic effects of HCMV; and simultaneously (2) treating the primary pathology (i.e., cancer).

Another consideration is the impact of altered metabolism on HCMV latency or reactivation. Herpesviruses reactivation remains broadly defined as a response to stress. Mitochondria constantly communicate with the nucleus, relaying information of metabolic and oxidative stress. Changes in cellular homeostasis are detected by the mitochondria and changes to cellular signaling or function can be initiated to reset homeostasis. Some changes are mediated through epigenetic remodeling, a change that has been shown to be mediated by metabolites. Recent literature suggests that epigenetic status or ROS may impact HCMV reactivation or latency. The effect of mitochondria-targeted antivirals on these pathways is an interesting question that deserves further study. As many of the prospective drugs described in this review decrease mitochondrial membrane potential and drive apoptosis pathways, it is also intriguing to consider the effect on long-term infection. Should these drugs induce apoptosis and prevent HCMV replication, the virus would in theory be unable to replicate and spread. Could this be a strategy to eliminate HCMV or other herpesviruses from our body? The effects of this or reduced reactivation may contribute to healthier aging. To translate the potential outlined in this review into clinical practice, a greater focus on viral mitochondrial dependency and pharmacological interventions must be undertaken.

## Figures and Tables

**Figure 1 viruses-15-01083-f001:**
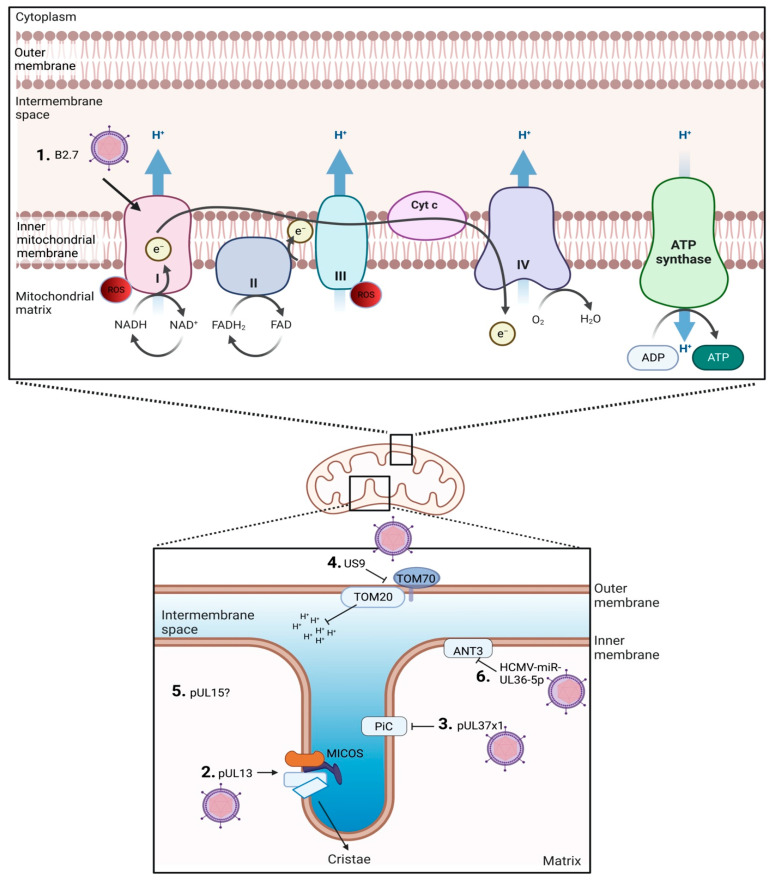
Overview of HCMV gene products associated with altering host mitochondria function. 1. lncβ2.7 interacts with ETC complex I. 2. pUL13 stabilizes the MICOS complex. 3. pUL37x1 inhibits PiC. 4. US9 disrupts TOM. 5. pUL15 colocalizes with mitochondria but the specific target is unknown. 6. HCMV-miR-UL36-5p inhibits ANT.

**Table 1 viruses-15-01083-t001:** Effect on HCMV replication when altering ETC activity.

Drug	Mitochondrial Target	Result	Reference
Rotenone	ETC Complex I	1 log reduction in HCMV titer (TR stain) No change in HCMV titer (Towne strain) Significant cell death reported	[20]
		Decreased HCMV titer, increased oxidative stress, decreased ATP No change in cell death reported	[32]
Metformin	ETC Complex I	1 log reduction in HCMV titer 2.5 log reduction in HCMV titer (reduced glucose conditions) No change in cell death reported	[20]
Antimycin A	ETC Complex III	2 log reduction in HCMV titer (TR strain) 0.5 log reduction in HCMV titer (Towne strain) No change in cell death reported	[20]
Oligomycin	ATP Synthase	1 log reduction in HCMV titer (TR strain) No change in HCMV titer (Towne strain) No change in cell death reported	[20]
			[14]
FCCP	ETC	No significant change in HCMV titer No change in cell death reported	[20]
Chloramphenicol	ETC	Reduced HCMV titers Increased cell death reported	[14]

## Data Availability

A data included in this review is publicly available (23 April 2023).

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
