# Peer review of "Targeting the Host Mitochondria as a Novel Human Cytomegalovirus Antiviral Strategy"

_viruses, 2023, doi:10.3390/v15051083_

Round 1

Reviewer 1 Report

Herpesviruses are ubiquitous pathogens for which latency-reactivation cycles are a continued source of viral shedding and disease burden.  Nucleoside analog acyclovir and associated derivatives are therapeutically effective, however novel strategies are needed to circumvent viral resistance.  

Bachman and Zwezdaryk review the current literature of mitochondria inhibitors previously shown to suppress HCMV and discuss the repurposing of mitochondria anti-cancer drugs as a novel antiviral strategy.  This review effectively analyzes the seminal papers in the field.  Comments are provided to increase the clarity and readability of the text as well as including an introductory foundation for the biology of HCMV and associated pathologies.   

Specific comments:

 -Lines 24-62: The introduction is clearly laid out for the integral relationship between HCMV and the mitochondria.  However, there is no background information discussing the biology of HCMV, clinical outcome of infection, and prevalence within the population.  This information is important for those readers unfamiliar with HCMV and aids to support a context as to why the investigation into novel therapeutics is a significant priority.     

-Lines 64-90, 111-145, 227-247, 250-296: These paragraphs contain an overly abundance of information summarizing the literature.  To aid in clarity and readability it may be best that these large paragraphs be broken down into their individual topics (drugs, viral genes, etc) with particular emphasis on explaining the broader implications.

-This literature review encompasses research on acute/primary infection models.  Any thoughts as to the role the mitochondria plays and the impact of mitochondrion inhibitors/drugs has on HCMV latency and viral reactivation models?

-Lines 127, 131: Change symbol to “lncb2.7”.

-Lines 315-317: Reference(s) missing for “clinical evidence”.

Author Response

Thank you all 3 reviewers for their thoughtful critiques and suggestions. WE have incorporated all reviewer comments into a revised manuscript. Specific details are below. Thank you for helping to improve this manuscript.

Reviewer 1

Bachman and Zwezdaryk review the current literature of mitochondria inhibitors previously shown to suppress HCMV and discuss the repurposing of mitochondria anti-cancer drugs as a novel antiviral strategy.  This review effectively analyzes the seminal papers in the field.  Comments are provided to increase the clarity and readability of the text as well as including an introductory foundation for the biology of HCMV and associated pathologies.   

Specific comments:

  1. Lines 24-62: The introduction is clearly laid out for the integral relationship between HCMV and the mitochondria.  However, there is no background information discussing the biology of HCMV, clinical outcome of infection, and prevalence within the population.  This information is important for those readers unfamiliar with HCMV and aids to support a context as to why the investigation into novel therapeutics is a significant priority

Thank you for this suggestion. We have included a paragraph with a brief overview of HCMV biology (line 35-41).

  1. Lines 64-90, 111-145, 227-247, 250-296: These paragraphs contain an overly abundance of information summarizing the literature.  To aid in clarity and readability it may be best that these large paragraphs be broken down into their individual topics (drugs, viral genes, etc) with particular emphasis on explaining the broader implications.

We have divided all paragraphs mentioned above into smaller paragraphs as suggested (line 78-120, 137-181, 220-246, 248-268, 270-294).

  1. This literature review encompasses research on acute/primary infection models.  Any thoughts as to the role the mitochondria plays and the impact of mitochondrion inhibitors/drugs has on HCMV latency and viral reactivation models?

Thank you for the wonderful question! We have included a paragraph in the discussion section (line 386-400)

  1. Lines 127, 131: Change symbol to “lncb2.7”.

Both lines have been corrected (line 155 and 158).

  1. Lines 315-317: Reference(s) missing for “clinical evidence”.

References have been added (line 381).

Reviewer 2 Report

This review article by Bachman and Zwezdaryk explores the potential of targeting host mitochondria function as novel anti-HCMV therapeutic strategy. This is an important area of research as current antivirals target viral proteins and are prone to viral resistance as well as high toxicity. The authors nicely summarize where the field currently lies in terms of how HCMV modulates mitochondria function to promote viral replication. However, the review does not discuss mitochondrial associated cell death as it relates to HCMV infection. Given regulation of cell death is a major function of the mitochondria, a more concise discussion about how targeting the mitochondria could affect both viral replication and/or the viability of HCMV-infected cells should be included. See additional comments below. 

1.    The authors state that this review will not discuss mitochondrial-associated apoptosis. However, targeting the mitochondria as an antiviral strategy will inevitably affect apoptosis and should be discussed, at least to the extent that it relates to HCMV infection. Particularly, given the extensive discussion about how mitochondrially targeted drugs for cancer all stimulate cell death. Targeting the mitochondria could have the benefit of limiting replication while also eliminated infected cells.

2.    There should be clearer discussion about how the drugs listed in Table 1 also affect cell viability. It is unclear as to whether the listed drugs directly attenuate HCMV replication, simply decrease cell viability, or both. This information should be incorporated into Table 1 as well.

3.    The authors state that developing new antivirals against HCMV is important in part due to the toxicity of current antivirals. However, there is no discussion about the toxicity of mitochondria targeting drugs used as cancer therapies.

Author Response

Thank you all 3 reviewers for their thoughtful critiques and suggestions. WE have incorporated all reviewer comments into a revised manuscript. Specific details are below. Thank you for helping to improve this manuscript.

Reviewer 2

This review article by Bachman and Zwezdaryk explores the potential of targeting host mitochondria function as novel anti-HCMV therapeutic strategy. This is an important area of research as current antivirals target viral proteins and are prone to viral resistance as well as high toxicity. The authors nicely summarize where the field currently lies in terms of how HCMV modulates mitochondria function to promote viral replication. However, the review does not discuss mitochondrial associated cell death as it relates to HCMV infection. Given regulation of cell death is a major function of the mitochondria, a more concise discussion about how targeting the mitochondria could affect both viral replication and/or the viability of HCMV-infected cells should be included. See additional comments below. 

  1. The authors state that this review will not discuss mitochondrial-associated apoptosis. However, targeting the mitochondria as an antiviral strategy will inevitably affect apoptosis and should be discussed, at least to the extent that it relates to HCMV infection. Particularly, given the extensive discussion about how mitochondrially targeted drugs for cancer all stimulate cell death. Targeting the mitochondria could have the benefit of limiting replication while also eliminated infected cells.

Your suggestion makes complete sense and the question posed at the end is wonderful. We have included a section on apoptosis in the introduction (line 56-62) and discuss the impact of replication in the discussion (386-400)

  1. There should be clearer discussion about how the drugs listed in Table 1 also affect cell viability. It is unclear as to whether the listed drugs directly attenuate HCMV replication, simply decrease cell viability, or both. This information should be incorporated into Table 1 as well.

Thank you for this comment. Clarification has been provided  (line 154-155, 178-180, 202-203, 245-246, and Table 1)

  1. The authors state that developing new antivirals against HCMV is important in part due to the toxicity of current antivirals. However, there is no discussion about the toxicity of mitochondria targeting drugs used as cancer therapies.

Great comment. We have addressed this by including a paragraph describing toxicity adverse events in clinical trial settings (line 348-352)

Reviewer 3 Report

Mitochondria have traditionally been considered as the energy plants of host cells; however, recent studies have demonstrated that they play active roles as regulators of cellular events, including interaction with invading viruses. This review explores the targeting of mitochondrial functions as a novel strategy for anti-viral infection, summarizing current knowledge on the regulation of mitochondrial functions by hCMV and interventions in viral infection by targeting mitochondria. However, the article's structure could be improved as it currently contains multiple paragraphs elaborating on both hCMV dysregulation of mitochondrial and the roles of mitochondria in hCMV infection, which may be confusing for readers. For instance, the description of hCMV lnc-beta2.7 in lines 128-129 is not clearly linked to either topic. To address this issue, the article could be restructured with separate sections addressing each topic. Additionally, Table 1 provides a comprehensive summary of compounds targeting electron transport chain (ETC) and inhibiting hCMV replication. Although the authors detail different compounds in subsections under section 3, not every mitochondrial inhibitor listed has been proved to have anti-hCMV effects. Therefore, creating a separate section describing the perspective of anti-hCMV by targeting ETC is suggested to avoid confusion. Lastly, the discussion of the anti-tumor effect of mitochondrial inhibition in section 5 is out of context and does not have any clear connection to anti-hCMV effects. As such, it is suggested to remove or relocate this section to improve the article's flow and coherence.

Author Response

Thank you all 3 reviewers for their thoughtful critiques and suggestions. WE have incorporated all reviewer comments into a revised manuscript. Specific details are below. Thank you for helping to improve this manuscript.

Reviewer 3

Mitochondria have traditionally been considered as the energy plants of host cells; however, recent studies have demonstrated that they play active roles as regulators of cellular events, including interaction with invading viruses. This review explores the targeting of mitochondrial functions as a novel strategy for anti-viral infection, summarizing current knowledge on the regulation of mitochondrial functions by hCMV and interventions in viral infection by targeting mitochondria. However, the article's structure could be improved as it currently contains multiple paragraphs elaborating on both hCMV dysregulation of mitochondrial and the roles of mitochondria in hCMV infection, which may be confusing for readers.

  1. For instance, the description of hCMV lnc-beta2.7 in lines 128-129 is not clearly linked to either topic. To address this issue, the article could be restructured with separate sections addressing each topic. Additionally, Table 1 provides a comprehensive summary of compounds targeting electron transport chain (ETC) and inhibiting hCMV replication.

Appreciate the comment. We have reworded sections for clarity and changed the name of Figure 1. Paragraphs have been broken down as discussed above (Reviewer 1 comment #2) also line 152-159. 

  1. Although the authors detail different compounds in subsections under section 3, not every mitochondrial inhibitor listed has been proved to have anti-hCMV effects. Therefore, creating a separate section describing the perspective of anti-hCMV by targeting ETC is suggested to avoid confusion.

Thank you for this suggestion. We agree and have corrected text to definitely state when an experiment targeting the ETC using HCMV was reported or is speculative. (line190-191, 216-218) 

  1. Lastly, the discussion of the anti-tumor effect of mitochondrial inhibition in section 5 is out of context and does not have any clear connection to anti-hCMV effects. As such, it is suggested to remove or relocate this section to improve the article's flow and coherence.

Great suggestion. We have renamed the section and specified that these drugs have not been tested with HCMV but are candidates that could be repurposed for HCMV antiviral testing (line 296, 299-300)